# A Lightweight Hybrid Model with Location-Preserving ViT for Efficient Food Recognition

**DOI:** 10.3390/nu16020200

**Published:** 2024-01-08

**Authors:** Guorui Sheng, Weiqing Min, Xiangyi Zhu, Liang Xu, Qingshuo Sun, Yancun Yang, Lili Wang, Shuqiang Jiang

**Affiliations:** 1School of Information and Electrical Engineering, Ludong University, Yantai 264025, China; shengguorui@ldu.edu.cn (G.S.); zhuxiangyi@m.ldu.edu.cn (X.Z.); xuliang@m.ldu.edu.cn (L.X.); sunqingshuo@m.ldu.edu.cn (Q.S.); wanglili@ldu.edu.cn (L.W.); 2Key Laboratory of Intelligent Information Processing, Institute of Computing Technology, Chinese Academy of Sciences, Beijing 100190, China; minweiqing@ict.ac.cn (W.M.); sqjiang@ict.ac.cn (S.J.); 3School of Computer Science and Technology, University of Chinese Academy of Sciences, Beijing 100190, China

**Keywords:** food recognition, lightweight, global feature, ViT, nutrition management

## Abstract

Food-image recognition plays a pivotal role in intelligent nutrition management, and lightweight recognition methods based on deep learning are crucial for enabling mobile deployment. This capability empowers individuals to effectively manage their daily diet and nutrition using devices such as smartphones. In this study, we propose an Efficient Hybrid Food Recognition Net (EHFR–Net), a novel neural network that integrates Convolutional Neural Networks (CNN) and Vision Transformer (ViT). We find that in the context of food-image recognition tasks, while ViT demonstrates superiority in extracting global information, its approach of disregarding the initial spatial information hampers its efficacy. Therefore, we designed a ViT method termed Location-Preserving Vision Transformer (LP–ViT), which retains positional information during the global information extraction process. To ensure the lightweight nature of the model, we employ an inverted residual block on the CNN side to extract local features. Global and local features are seamlessly integrated by directly summing and concatenating the outputs from the convolutional and ViT structures, resulting in the creation of a unified Hybrid Block (HBlock) in a coherent manner. Moreover, we optimize the hierarchical layout of EHFR–Net to accommodate the unique characteristics of HBlock, effectively reducing the model size. Our extensive experiments on three well-known food image-recognition datasets demonstrate the superiority of our approach. For instance, on the ETHZ Food–101 dataset, our method achieves an outstanding recognition accuracy of 90.7%, which is 3.5% higher than the state-of-the-art ViT-based lightweight network MobileViTv2 (87.2%), which has an equivalent number of parameters and calculations.

## 1. Introduction

Food computing has gained increased attention in the fields of food and nutrition [1]. This is due to its potential to contribute to improvements in diet, health, and the food industry [2,3,4,5]. For example, by analyzing factors such as the meal type, components, and other distinctive information, one can evaluate the nutritional value of a meal and understand individual dietary habits. This not only ensures the person’s health but also aids in illness prevention [6]. Food-image recognition is of utmost importance to these application scenarios [7,8,9] Since the ultimate objective of the food computing system is to aid individuals in the management of their diet and health, as well as to enhance their daily activities, it becomes imperative to establish an efficient system specifically designed for the identification of food images on end devices, such as mobile phones [10]. In addition, the wide range of foods and cooking techniques has led to a rapid expansion of images of food, which has raised the expectation of a long-term expansion of image recognition on the server side. Lastly, the recognition of food images belongs to the more complex recognition of fine-grained objects [11], and the lightweight effort in this area will provide the same reference for fine-grained recognition models. However, current state-of-the-art techniques [12,13], predominantly leveraging deep-learning-based solutions, which entail a substantial number of parameters and involve an extensive process of training and evaluation. In light of this, this paper concentrates on the lightweight nature of deep neural network models for food image recognition.

At present, scant research has been undertaken on lightweight food-image recognition. Lightweight Convolutional Neural Network (CNN) was initially employed for food-image identification [10,14,15,16]. The primary challenge is that the traditional convolutional model is not capable of extracting long-range information from food images due to the dispersed arrangement of components. To gather more long-range features, a deeper CNN network is necessary, thus preventing the achievement of a lightweight model. As Figure 1 highlights, food images exhibit the characteristics of small inter-category differences and large intra-category differences. Key features in these images often rely on ingredient information scattered throughout the image. Moreover, the same component in the same dish can display different characteristics in terms of size, form, and dispersion based on the means of cooking, such as the two braised pork images in Figure 1. Hence, it is imperative to precisely capture the global features that represent distant correlations between these disparate food ingredients in order to identify the dish effectively. Vision Transformer (ViT) is adept at capturing global information and utilizes an attention strategy to identify aspects from various locations that are dependent on the input data. Unfortunately, this necessitates considerable computational expenses, and the training can be arduous. CNN focuses on modeling local relationships and has strong prior inductive biases. Sheng [17] tried to combine the local feature extraction ability of CNN and the global feature expression ability of ViT, but the parameters and calculation amount of the model are still quite large.

Therefore, the challenges of lightweight food-image recognition derive two-fold: (1) Due to the fine-grained characteristics and widespread, dispersed distribution of ingredients in food images, the extraction of global features becomes crucial for effective food-image recognition. CNN is good at perceiving local features and needs to build a highly complex network to obtain global features. This will elicit a sharp rise in the number of parameters and computations, which is incongruous with the demands of a lightweight network. (2) ViT proves to be an efficient way of acquiring features that reflect long-range pixel correlations. Unfortunately, the large number of token interactions necessitates a considerable amount of vector dot product calculations, as well as a larger pool of training data and iterations in order to obtain the global correlations. Therefore, to obtain efficient training on the server side and fast inferencing on the terminal like mobile phones while ensuring recognition accuracy, it is an urgent problem to be solved.

Our work has successfully tackled the key issues in lightweight food recognition, including the limited global information representation capacity of CNN and the complexity and difficulty of training the ViT model. We use LP–ViT to efficiently capture the global features in food images and form a skip connection-based series block (named HBlock) with the lightweight CNN-based module inverted residual block. HBlock is used as the basic structure of EHFR–Net, which effectively improves food-image recognition accuracy. Furthermore, based on the fact that LP–ViT focuses on extracting global features in the front part of EHFR–Net, we drastically reduce the number of network layers in the middle and rear parts to reduce the number of parameters and computations. We conduct comprehensive experiments on several major datasets in the food-image field. The results show that compared with existing CNN-based, ViT-based, and hybrid lightweight networks, in the case of equal or fewer parameters and FLOPs, our method has certain advantages in recognition accuracy.

We summarize our contributions as follows:A ViT module LP–ViT (Location-Preserving Vision Transformer) Block that effectively extracts global features of food images is designed and implemented. LP–ViT directly obtains the correlation between all pixels while maintaining the original spatial structure of the image through a series of reversible operations, thereby achieving more efficient fusion with the local features extracted by CNN.The HBlock (Hybrid Block) based on the series structure of LP–ViT block and CNN is designed and used as the backbone to establish a neural network Efficient Hybrid Food Recognition Net (EHFR–Net) that effectively recognizes food images.In view of the characteristics of the LP–ViT block, which starts to extract global features at the shallow layer of the network, an adapted neural network hierarchical layout structure is designed to effectively reduce the number of parameters and calculations and further achieve lightweight.We conduct extensive experiments on three open-source food image recognition datasets. Results demonstrate the effectiveness of our method, surpassing state-of-the-art CNN-based, ViT-based, and hybrid lightweight models with simple training recipes.

## 2. Related Works

### 2.1. Lightweight CNNs, ViTs, and Hybrid Models

ResNet [18] is one of the most renowned CNN architectures. Nevertheless, the most efficacious CNN models typically require a high number of parameters and FLOPs. Lightweight CNNs that achieve competitive performance with a reduced number of parameters and FLOPs comprise ShuffleNetv2 [19], ESPNetV2 [20], EfficientNet [21], MobileNetV2 [22], and V3 [23]. MobileNetV3 [23] is the most current implementation of a set of models tailored for situations with limited resources, like mobile devices. The basic blocks of MobileNetV3 [23] include the MobileNetV2 [22] block and the Squeeze-and-Excite network [24]. The common complication of CNN-based lightweight architectures is their deficiency of capacity to extract global features.

For the purpose of rapidly obtaining global information, ViT [25] brings transformer models tailored to natural language processing tasks to the vision domain, particularly image recognition. The incorporation of ViT into machine vision has piqued scholarly interest in its potential for lightweightness. A majority of efforts are being channeled towards perfecting the self-attention procedure to enhance productivity, e.g., SwinT [26], EfficientFormer [27], LightViT [28], EfficientViT [29], MiniViT [30], and TinyViT [31]. Training difficulties and the exorbitant computational expense resulting from the quadratic number of interactions between tokens are common predicaments of ViT-based lightweight models.

Recently, researchers have endeavored to assemble compact hybrid systems that integrate CNN and ViT for mobile vision tasks, which indicates that amalgamating convolution and transformer yields enhancement in prediction precision, as well as training dependability. Subsequently, there have been a large number of lightweight works on these models, such as MobileFormer [32], CMT [33], CvT [34], BoTNet [35], Next–ViT [36], EdgeViTs [37], MobileViTv1 [38], and MobileViTv2 [39]. The hybrid lightweight model structuring CNN and ViT has realized an impressive combination of global and local information, although the issue of a voluminous model remains.

### 2.2. Lightweight Food Recognition

Recently, Min et al. [1] delivered a comprehensive study on food computing, which encompassed food recognition. In the initial stages, many manually crafted features are utilized for recognition [40,41]. For instance, Lukas et al. [41] applied random forests to extract distinctive image patches as a visual representation. With the growth of deep-learning technology, numerous recognition methods based on deep learning have been developed [12,13,42,43,44,45,46].

Due to the necessity of lightweight food-image recognition, a lot of related research work has been proposed. Early researchers used the lightweight CNN method for food-image recognition [10,14,15,16]. Tan et al. [16] recently proposed a novel lightweight Neural Architecture Search (LNAS) model to self-generate a thin CNN that can be executed on mobile devices, achieving nearly 76% recognition accuracy on the ETHZ Food–101 dataset. The efficacy of these CNN-based lightweight food-recognition models is not especially high. ViT presents a new method for obtaining global information on food images, Sheng et al. [17] tried to extract global and local features with a parallel structure composed of the ViT group and CNN, and it obtained the SOTA performance. However, due to the Multi-Head Attention mechanism of the ViT, the model size is still large.

## 3. Materials and Methods

### 3.1. Datasets and Training Settings

In order to assess the proposed model, we conduct experiments on three food datasets: ETHZ Food–101 [41], Vireo Food–172 [47], and UEC Food256 [48]. ETHZ Food–101 is the first large-scale Western cuisine image dataset, comprising 101 classes of Western dishes with a total of 101,000 image. We used 75,750 images for training and 25,250 for validation. Vireo Food–172, released in 2016, is the first relatively large-scale dataset of Chinese cuisine, featuring 172 classes and a total of 110,241 food images. We used 66,071 images for training and 44,170 images to validate. UEC Food256 exclusively comprises Japanese dishes, with a total of 25,088 images spanning across 256 food categories. Among them, 22,095 images were allocated for training and 9300 for validation.

We train our models using an input image resolution 256×256, a batch size of 32, and AdamW [49] optimizer. We linearly increase the learning rate from $10^{-6}$ for the first 20 k iterations, and then a cosine schedule with a learning rate ranging from 0.0002 to 0.002. Furthermore, we use the same data augmentation method as MobileViTv2 [39] for image preprocessing.

### 3.2. Overview of EHFR–Net

EHFR–Net is a hybrid neural network based on CNN and ViT. Since a significant feature of food images is that different dishes may have the same ingredients, this will lead to the inability to obtain accurate results when using local features as the main discriminant factor for food-image recognition. Therefore, effective extraction of global features is particularly important in food-image recognition. Since ViT has more prominent global feature extraction capabilities than CNN, a natural idea is to effectively combine convolution and ViT: use convolution to extract local features and ViT to obtain global features to improve the model’s expression ability in food image recognition tasks. Unlike other hybrid networks that combine CNN and ViT, we designed a LP–ViT Block that can retain the original spatial information of the image while calculating the correlation between all pixels. This structure is more effective in dealing with food images and is more lightweight.

Figure 2a is the overall architecture diagram of EHFR–Net and its main parts. Figure 2b shows that HBlock is implemented by CNN and LP–ViT block in series, and the local features and global features are effectively integrated into HBlock. Figure 2c illustrates the implementation method of LP–ViT. After unfolding the input, it traverses the parallel separate self-attention module and patch attention module. Subsequently, it proceeds through an FFN layer and finally reinstates the position information through the fold operation. Figure 2d shows the implementation of the transposition operation in the patch attention module and the extraction of full-pixel-related features inside patches. Figure 2e shows the image processing process of HBlock.

### 3.3. LP–ViT

The traditional vision transformer needs to convert the input into a token sequence through the embedding layer and then calculate the correlation. The embedding layer compresses all the pixels in a patch into one pixel through convolution and then unfolds it into a sequence through a flattening operation. However, since the irreversibility of the embedding operation will destroy the original spatial information of the image itself, positional encoding is needed to generate separate location information for each token, as shown in Figure 3. LP–ViT replaces the embedding layer and positional encoding in traditional ViT with the help of unfold and fold operations, so that the image transformed into a sequence can be restored back to the image. This greatly retains the original spatial information of the image and is more conducive to the recognition of food images. In addition, we use separate self-attention in LP–ViT, which is more lightweight and has a smaller latency than Multi-Head Attention.

The specific processing is shown in Figure 4. The input is first unfolded to form a sequence of pixels at the same position in each patch, as shown in Figure 4a, and then sent to the left branch of the parallel structure in Figure 2c to calculate the correlation. At the same time, the generated sequence is sent to the patch attention of the right branch to calculate the correlation of pixels within the patch. In patch attention, as depicted in Figure 4b, the sequence obtained by unfolding is utilized to arrange the pixels inside the patch into a sequence via the transpose operation. Subsequently, the correlation is calculated through the attention structure to obtain the output, which is then restored through transpose. The outputs from the left branch and the right branch are fused by addition and sent to FFN to obtain the output. After that, the sequence is restored back to the image arrangement form through fold operation. After processing by the LP–ViT module, the correlation between all pixels in the image is obtained, as shown in Figure 4c. In this way, in the food-image recognition task, the global features can be locked more quickly and accurately, and the model expression effect is better.

The following takes a batch as an example to introduce the detailed processing process of LP–ViT. The patch size is 2×2, and the input is a matrix:(1)$$\left(\begin{array}{ccc} x_{11}^{1} & \ldots & x_{1w}^{1}\\ \vdots & \ddots & \vdots \\ x_{h1}^{1} & \cdots & x_{hw}^{1} \end{array}\right)\cdots \left(\begin{array}{ccc} x_{11}^{c} & \ldots & x_{1w}^{c}\\ \vdots & \ddots & \vdots \\ x_{h1}^{c} & \cdots & x_{hw}^{c} \end{array}\right)\in R^{1\times C\times H\times W}$$

Input X is obtained through unfolding and reshaping operations:
(2)$$\begin{array}{l} \left(\begin{array}{ccc} x_{11}^{1} & \cdots & x_{1w}^{1}\\ \vdots & \ddots & \vdots \\ x_{h1}^{1} & \cdots & x_{hw}^{1} \end{array}\right)\cdots \left(\begin{array}{ccc} x_{11}^{c} & \cdots & x_{1w}^{c}\\ \vdots & \ddots & \vdots \\ x_{h1}^{c} & \cdots & x_{hw}^{c} \end{array}\right)\overset{\mathrm{Unfold}}{\to }\left(\begin{array}{ccc} X_{2m-1,2n-1}^{1} & \cdots & X_{2m-1,2n-1}^{C}\\ X_{2m-1,2n}^{1} & & X_{2m-1,2n}^{C}\\ X_{2m,2n-1}^{1} & & X_{2m,2n-1}^{C}\\ X_{2m,2n}^{1} & \cdots & X_{2m,2n}^{C} \end{array}\right)\\ \overset{\mathrm{Reshape}}{\to }\left(\begin{array}{c} X_{2m-1,2n-1}^{1}\\ X_{2m-1,2n}^{1}\\ X_{2m,2n-1}^{1}\\ X_{2m,2n}^{1} \end{array}\right)\cdots \left(\begin{array}{c} X_{2m-1,2n-1}^{C}\\ X_{2m-1,2n}^{C}\\ X_{2m,2n-1}^{C}\\ X_{2m,2n}^{C} \end{array}\right)=X \end{array}$$
where $X_{2m-1,2n-1}^{C}$ represents $x_{ij}^{c}$ with odd row and column coordinates in the C−th channel, $X_{2m-1,2n}^{C}$ represents $x_{ij}^{c}$ with odd row coordinates and even column coordinates in the C−th channel, $X_{2m,2n-1}^{C}$ represents $x_{ij}^{c}$ with even row coordinates and odd column coordinates in the $X_{2m,2n}^{C}$ represents $x_{ij}^{c}$ with even row and column coordinates in the C−th channel. $X\in R^{1\times C\times P\times N}$, P=2×2, $N=\frac{H\times W}{P}$.

Then, X enters the left branch to perform the separate self-attention operation:(3)$$Y_{1}=\left\{\sum \left(\sigma (XW_{I})*XW_{K})*\mathrm{ReLU}(XW_{V}\right)\right\}W_{O}$$
where $W_{I}\in R^{N\times 1}$, $W_{K}\in R^{N\times N}$, $W_{V}\in R^{N\times N}$, $W_{O}\in R^{N\times N}$, $Y_{1}\in R^{1\times C\times P\times N}$, σ means Softmax activation function, ∗ and ∑ are broadcastable element-wise multiplication and summation operations, respectively. Perform patch attention in the right branch:(4)$$Y_{2}={\left\{\left[\sum \left(\sigma (X^{T}W_{I}^{\prime })*X^{T}W_{K}^{\prime })*\mathrm{ReLU}(X^{T}W_{V}^{\prime }\right)\right]W_{O}^{\prime }\right\}}^{T}$$
where $W_{I}^{\prime }\in R^{P\times 1}$, $W_{K}^{\prime }\in R^{P\times P}$, $W_{V}^{\prime }\in R^{P\times P}$, $W_{O}^{\prime }\in R^{P\times P}$,$Y_{2}\in R^{1\times C\times P\times N}$. Add the results from the two branches and perform the FFN operation to obtain the output Y:(5)$$Y=W_{2}\left[W_{1}\left(Y_{1}+Y_{2}\right)+B_{1}\right]+B_{2}$$
where $W_{1}\in R^{1\times 2C\times H\times W}$, $W_{2}\in R^{1\times C\times H\times W}$, $B_{1}\in R^{1\times 2C\times H\times W}$, $B_{2}\in R^{1\times C\times H\times W}$, $Y\in R^{1\times C\times P\times N}$. Finally, Y obtains the output through the fold operation. The fold operation is essentially the reverse process of the unfold operation:(6)$$Output=\left(\begin{array}{ccc} y_{11}^{1} & \ldots & y_{1w}^{1}\\ \vdots & \ddots & \vdots \\ y_{h1}^{1} & \cdots & y_{hw}^{1} \end{array}\right)\cdots \left(\begin{array}{ccc} y_{11}^{c} & \ldots & y_{1w}^{c}\\ \vdots & \ddots & \vdots \\ y_{h1}^{c} & \cdots & y_{hw}^{c} \end{array}\right)\in R^{1\times C\times H\times W}$$

### 3.4. HBlock and Overall Network Architecture

#### 3.4.1. HBlock

HBlock is composed of a combination of convolution and ViT, which serves as the basic module that constitutes the backbone of the entire EHFR–Net model. Among them, the convolution part uses the Inverted Residual Block proposed in MobileNetV2, and the ViT part uses the LP–ViT module proposed in this article. In HBlock, the convolution extracts local features of food images, and LP–ViT encodes global information on the local features extracted by the convolution so that the correlation information among all pixels in the feature map can be extracted. Then, local features are fused with global information through skip connection. The design method of HBlock can enable the model to quickly lock the characteristics of the ingredients in the picture and the global characteristics of the food image, improving the accuracy of the model.

#### 3.4.2. Overall Structure of EHFR–Net

In order to make the model more lightweight while ensuring accuracy, we redesigned a new network hierarchical layout that is different from the traditional network structure. This is mainly based on the following considerations: First, since food-image recognition is a fine-grained image recognition task, and convolution can extract fine-grained features of the image in the shallow network, we increased the number of Inverted Residual Blocks in the shallow network. Second, traditional convolutional networks better extract the global information of the image by increasing the number of modules in the deep network, while the basic module HBlock of EHFR–Net is composed of a combination of convolution and ViT, allowing the model to start extracting global features in the shallow network. Therefore, we reduce the number of Inverted Residual Blocks in the deep network. Third, although the shallow network of EHFR–Net focuses more on the extraction of fine-grained features, the existence of LP–ViT also extracts some global information. On the other side, both convolution operations and LP–ViT in deep networks can extract global information, so global information is acquired throughout the entire network. In response to this feature, we use fewer LP–ViT blocks in the shallow and deep layers of the entire network structure and relatively more in the middle layer. Based on the above three adjustments to the network structure, our model is more lightweight, more suitable for food-image recognition tasks, and it has stronger model-expression capabilities. The specific network structure is shown in Table 1.

## 4. Results

### 4.1. Results on ETHZ Food–101

Results from ETHZ Food–101 are displayed in Table 2. The results have been divided into groups based on a similar set of parameters. Our model is better than all others in six parameter ranges. Among all models with less than 1 M parameters, our model achieves 89.4% top-1 accuracy, which is 15% higher than ShuffleNet V2 [19]. Although it incurs higher computational costs, the recognition accuracy has significantly improved. Due to the rapid development of end-device hardware in today’s context, the computational power required for this level of processing is entirely acceptable. Among all models with 1 M–2 M parameters, our model achieves 90.4% top-1 accuracy, which is 9.2%, 5.2%, and 3.5% higher than GhostNetV2 [50], MobileNetV3 [23], MobileViTv2 [39], and MobileNetV2 [22], respectively. In around 2–3 M, 4–5 M, and 6–10 M parameter budget models, our model’s top-1 accuracy is 90.7%, 91.1%, and 91.3%, which is at least 4% higher than the accuracy achieved by the current mainstream lightweight CNN-based models such as MobileNetV3 [23], GhostNetV2 [50], and EfficientNet [21], and at least 3% higher than ViT-based models such as MobileViTv2 [39]. In the parameter size range from 10 to 20 M, our model achieves the highest recognition accuracy with the smallest number of parameters and has lower FLOPs than mobileViTv2 [39]. Compared with the CNN-based model GhostNetV2 [50], the performance is improved by nearly 6%. We also compared the performance with recent lightweight food recognition networks. The results show that the recognition accuracy of our network (90.7%) is much higher than that of LNAS–NET [16] and LTBDNN(TD–192) [17] (76.8%), in the case of a comparable or significantly fewer number of parameters.

### 4.2. Results on Vireo Food–172

The results for VireoFood–172 are shown in Table 3. Our method achieves optimal performance under various ranges of parameter quantities, improving performance by nearly 5% compared with the CNN-based model MobileNetV3 [23]. Compared with the ViT-based SOTA model MobileViTv2 [39], our method improves the performance by 3%. We have concluded that the superior performance on Vireo Food–172 is attributable to the dataset containing a larger variety of Chinese dishes, as well as having a greater range of ingredients.

### 4.3. Results on UEC Food256

Table 4 reveals results that are similar to those of the other two datasets. Our method achieves optimal performance under various ranges of parameter quantities, improving performance by more than 5% compared with the CNN-based model MobileNetV3 [23]. Although the FLOPs of the method are higher than those based on CNN, the significant improvement in recognition accuracy is worthwhile. Compared with the ViT-based SOTA model MobileViTv2 [39], our method improves the performance by 1–2% with fewer calculations.

### 4.4. Qualitative Analysis and Visualization

Unlike traditional CNN-based models that focus on local features, the EHFR–Net shows better quality in extracting global features. Figure 5 shows the comparison by the method provided by Grad–CAM [51]. Results are obtained using only local convolution and using both the LP–ViT-based global feature and the CNN-based local feature. In Figure 5, (a) is from dataset ETHZ Food–101; and (b) is from Vireo Food–172. The three columns on the left are examples of food-image types that can be correctly identified by both methods, and the seven columns on the right are examples that only EHFR–Net can correctly identify but the CNN-based method failed. The first row is the original image, the second is heat maps generated using only local convolution, and the third is heat maps generated by EHFR–Net. As can be seen from Figure 5: (1) For food images with single ingredients and simple backgrounds, both methods can achieve accurate recognition by accurately focusing on local features of the image; (2) For food images with diverse ingredients and complex backgrounds, traditional CNN-based networks often cannot focus on key distinctive features and are more susceptible to background interference. EHFR–Net is better able to capture global information and key distinctive information; (3) Since Chinese food has richer ingredients, the diversity and complexity of food images are higher, and the requirements for the model’s ability to grasp global information is higher. The experimental results of EHFR–Net on the dataset Vireo Food–172 show that it has significantly stronger capabilities in this regard. In summary above, results show that the EHFR–Net is more suitable for food images and can achieve better recognition results.

### 4.5. Ablation Study

In this section, we ablate important design elements in the proposed model EHFR–Net using image classifications on three datasets. The results are summarized in Table 5.

Effectiveness of LP–ViT. We first present an ablation study to verify the efficiency of the proposed LP–ViT design by replacing the LP-ViT block with the original Multi-Head Attention (MHA) block. Compared with the ViT model using the ordinary MHA, the EHFR–Net using LP–ViT achieved the highest Top-1 accuracy with significantly reduced calculation and fewer parameters: 90.67% versus 88.80% (Food–101), 91.38% versus 89.66% (Food–172), 71.30% versus 69.28% (Food256). The computational effort is only about one-sixth that of the MHA. This shows that LP–ViT can extract global features from food images more efficiently, thereby improving the accuracy and efficiency of the model.Effectiveness of HBlock integrated with CNN and LP–ViT. We designed a module HBlock that combines the CNN and ViT in series as the basic module of the model. Compared with models that only use CNN and models that only use ViT, EHFR–Net’s Top-1 recognition rate has significant advantages: 90.67% versus 88.37%, 87.81% (Food–101), 91.38% versus 90.05%, 89.86% (Food–172), 71.30% versus 68.27%, 68.19% (Food256). The results show that the fusion CNN and ViT strategy designed by HBlock based on the characteristics of food images can effectively extract local and global features to achieve better recognition results.Effectiveness of adjusted network architecture. Based on the characteristics of the food-image recognition task, we designed a new architecture that is different from the traditional hybrid model network structure. This structure allows our model to achieve higher accuracy while further achieving lightweight. Compared with networks using the same modules but traditional structures, EHFR–Net achieves higher accuracy in a smaller size: (2.82 M, 90.67%) versus (4.52 M, 87.95%) (Food–101), (2.84 M, 91.38%) versus (4.55 M, 89.36%) (Food–172), (2.87 M, 71.30%) versus (4.57 M, 67.26%) (Food256).

### 4.6. Discussion

In the realm of lightweight neural networks, such as ShuffleNetv2 [19], ESPNetV2 [20], EfficientNet [21], MobileNetV2 [22], and MobileNetV3 [23], notable advancements have been achieved in terms of lightweight design. However, their performance in food-image recognition tasks falls short due to their limitation in extracting global information from shallow networks, relying solely on convolutions for local feature extraction. On the other hand, pure Vision Transformer (ViT) networks, exemplified by SwinT [26], EfficientFormer [27], LightViT [28], EfficientViT [29], MiniViT [30], and TinyViT [31], excel in capturing global information but struggle to effectively capture local fine-grained features of images. Moreover, the inherent characteristics of transformer structures contribute to larger parameter and computational requirements, compromising the balance between high-recognition accuracy and lightweight design. Hybrid networks that combine convolutional and ViT architectures, such as MobileFormer [32], CMT [33], CvT [34], BoTNet [35], Next–ViT [36], EdgeViTs [37], MobileViTv1 [38], and MobileViTv2 [39], face challenges in achieving an efficient fusion of local and global features, resulting in limited accuracy gains. Additionally, the use of traditional network structures hinders the extent of model lightweighting.

Our proposed EHFR–Net model successfully addresses these challenges by achieving an efficient fusion of convolutional and ViT structures, enabling the effective extraction of both local features and global information. The adoption of a novel network architecture, distinct from traditional structures, allows our model to achieve higher accuracy with reduced parameters and computational overhead.

However, our model, with its predominant use of ViT structures in the overall architecture and the utilization of the original model parameters during the inference stage, introduces latency concerns. Future work will focus on optimizing latency by implementing reparameterization techniques during the inference stage, aligning the model for better applicability on mobile devices. This optimization aims to contribute more effectively to daily dietary and nutritional management.

## 5. Conclusions

This work proposes a lightweight hybrid neural network model named EHFR–Net, which integrates CNN and ViT in order to capture the specific characteristics of food images. On the ViT side, we use the LP–ViT module to allow all pixels to participate in correlation calculations and ensure that the position information between patches is not destroyed through a series of reversible operations, thereby achieving efficient global feature extraction. The global features extracted by the LP–ViT module are integrated with the lightweight CNN network inverted residual module through a skip connection to achieve more accurate food-image recognition. We adopted a lightweight network design on both the ViT side and the CNN side and adjusted the overall network layout of EHFR–Net based on the characteristics of LP–ViT, thus achieving a lightweight overall model.

Lightweight food-image recognition is more conducive to deployment on terminal devices such as mobile phones, providing users with convenient and comprehensive nutritional information, effectively serving people’s health management. Future work includes research on implementing lightweight neural network models containing food image recognition and recommendation functions that match specific hardware and operating systems.

## Figures and Tables

**Figure 1 nutrients-16-00200-f001:**
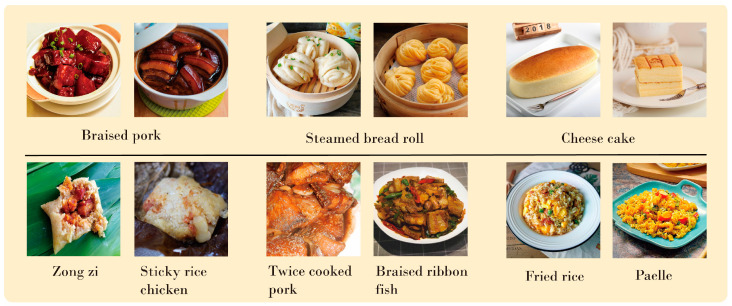
Examples from ETHZ Food–101 and Vireo Food–172. The first row of samples shows that food images have the characteristics of large intra-class differences; the second row of samples shows that food images have the characteristics of small inter-class differences.

**Figure 2 nutrients-16-00200-f002:**
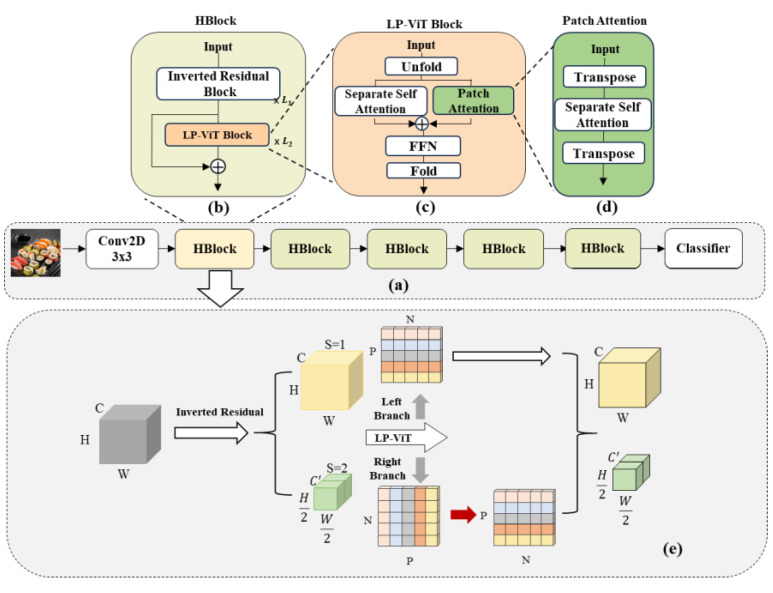
The overall architecture of EHFR–Net. (**a**) The backbone of EHFR–Net. It is composed of concatenated HBlocks. (**b**) The architecture of HBlock. L1 and L2 denote the number of blocks. HBblock comprises Inverted Residual Blocks capable of extracting local features and LP–ViT Blocks efficient in extracting global information. The local features and global features are further fused through skip connections. (**c**) The architecture of LP–ViT Block. It utilizes the unfold operation to expand pixel points into a sequence. These sequences are then separately fed into the left and right branches of the attention structure to compute the correlations between pixel points within patches and between patches. Subsequently, global information is obtained, and finally, the fold operation is applied to restore the sequence back to the image shape, achieving a local-preserving effect. (**d**) The architecture of Patch Attention. Computes the correlations between pixel points within patches through a combination of transpose and attention operations. (**e**) Image-processing process of HBlock. Initially, the image undergoes local feature extraction through the Inverted Residual Block, simultaneously considering downsampling operations. The obtained feature map is then fed into the LP–ViT Block, where two branches independently calculate the correlations between pixels within patches and between patches. These correlations are later fused to acquire global information.

**Figure 3 nutrients-16-00200-f003:**
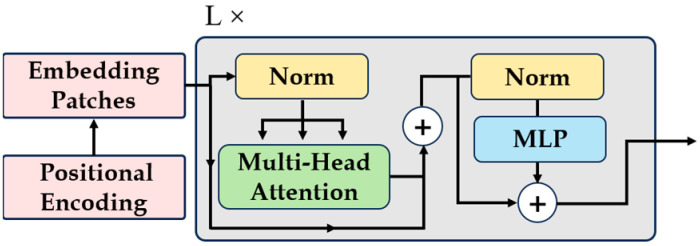
Initial processing of traditional ViT. The input is converted into a token sequence through the embedding layer and then the correlation is calculated. Positional encoding is required to generate separate position information for each token. This is because the embedding operation is irreversible and destroys the original spatial information of the image itself.

**Figure 4 nutrients-16-00200-f004:**
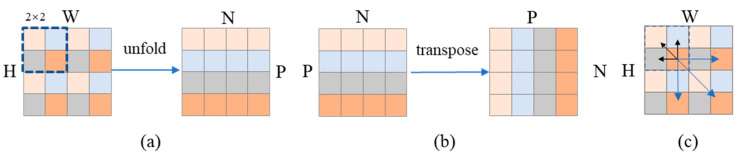
Processing of LP–ViT. (**a**) The input is first unfolded to form a sequence of pixels at the same position in each patch; (**b**) Arrange the pixels inside the patch into a sequence through the transpose operation; (**c**) LP–ViT extracts more comprehensive global features.

**Figure 5 nutrients-16-00200-f005:**
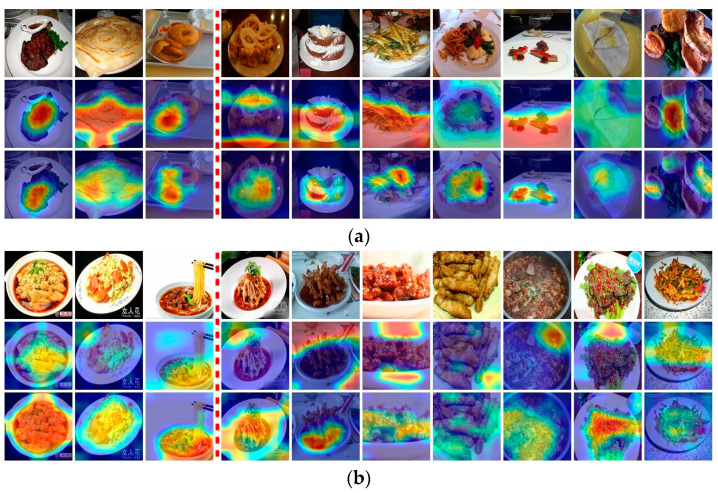
Visualization of experimental results comparison. (**a**): Examples from dataset Food–101 (**b**): Examples from dataset Vireo Food–172. The first row is the original image, the second row is the heat map without LP–ViT block, and the third row is the heat map with LP–ViT block. The first three columns both have achieved good recognition results, while in the last seven columns, only EHFR–Net is correct, and pure CNN-based method fails.

**Table 1 nutrients-16-00200-t001:** Network specification. Exp Ratio: Expansion Ratio in MobileNetV2 [22] block; α∈0.5,2.0: width multiplier to create models at different complexities of EHFR–Net.

Component	Input	Operator	Exp Ratio	Patch Size	Output Channel	Stride
Head	256×256	Conv2D 3×3	-	-	32α	2
Block Group 1	128×128	Inverted Residual	1	-	32α	2
	128×128	Inverted Residual	1	-	32α	1
	128×128	LP-ViT	-	2×2	32α	-
Block Group 2	128×128	Inverted Residual	1	-	64α	2
	64×64	Inverted Residual	1	-	64α	1
	64×64	Inverted Residual	3	-	64α	1
	64×64	LP-ViT	-	2×2	64α	-
	64×64	LP-ViT	-	2×2	64α	-
Block Group 3	64×64	Inverted Residual	3	-	96α	2
	32×32	Inverted Residual	3	-	96α	1
	32×32	Inverted Residual	3	-	96α	1
	32×32	LP-ViT	-	2×2	96α	-
	32×32	LP-ViT	-	2×2	96α	-
	32×32	LP-ViT	-	2×2	96α	-
Block Group 4	32×32	Inverted Residual	6	-	160α	2
	16×16	Inverted Residual	2.5	-	160α	1
	16×16	Inverted Residual	2.5	-	160α	1
	16×16	LP-ViT	-	2×2	160α	-
	16×16	LP-ViT	-	2×2	160α	-
Block Group 5	16×16	Inverted Residual	6	-	320α	2
	8×8	LP-ViT	-	2×2	320α	-

**Table 2 nutrients-16-00200-t002:** Performance comparison on ETHZ Food–101. EHFR–Net–x: x denotes the width multiplier on the base model.

Method	Top-1 Acc.	#Params	#FLOPs
ShuffleNetV2-0.5 [19]	74.3%	0.5 M	41.6 M
**EHFR** **–Net-0.5**	**89.4%**	0.8 M	428.3 M
MobileViTv2-0.5 [39]	87.0%	1.1 M	480.2 M
ShuffleNetV2-1.0 [19]	78.0%	1.4 M	148.8 M
MobileNetV3-0.5 [23]	82.4%	1.5 M	73.3 M
GhostNetV2-0.5 [50]	81.2%	1.7 M	54.0 M
**EHFR** **–Net-0.75**	**90.4%**	1.8 M	981.9 M
MobileViTv2-0.75 [39]	87.2%	2.5 M	1051.4 M
ShuffleNetV2-1.5 [19]	80.3%	2.6 M	303.6 M
MobileNetV3-0.75 [23]	85.5%	2.8 M	161.9 M
**EHFR** **–Net-1.0**	**90.7%**	2.8 M	1238.5 M
MobileNetV3-1.0 [23]	86.2%	4.3 M	218.9 M
MobileViTv2-1.0 [39]	87.6%	4.4 M	1843.4 M
**EHFR** **–Net-1.25**	**91.1%**	4.5 M	2104.5 M
EfficientNeT B0 [21]	85.2%	4.7 M	566.9 M
GhostNetV2-1.0 [50]	83.6%	5.0 M	176.9 M
ShuffleNetV2-2.0 [19]	82.0%	5.6 M	596.4 M
MobileNetV3-1.25 [23]	86.2%	6.4 M	366.8 M
**EHFR** **–Net-1.0**	**91.3%**	6.4 M	2985.5 M
MobileViTv2-1.25 [39]	88.3%	6.9 M	2856.0 M
GhostNetV2-1.3 [50]	84.8%	7.8 M	282.5 M
MobileNetV3-1.5 [23]	86.5%	8.6 M	500.4 M
MobileViTv2-1.5 [39]	88.6%	9.9 M	4089.3 M
**EHFR** **–Net-2.0**	**91.5%**	11.1 M	4787.4 M
GhostNetV2-1.6 [50]	85.5%	11.2 M	415.0 M
MobileViTv2-1.75 [39]	88.9%	13.4 M	5543.5 M
GhostNetV2-1.9 [50]	85.7%	15.3 M	572.8 M
MobileViTv2-2.0 [39]	89.5%	17.5 M	7218.3 M
LNAS–NET [16]	75.9%	1.8 M	-
LTBDNN(TD–192) [17]	76.8%	12.2 M	-
**EHFR** **–Net-1.0**	**90.7%**	2.8 M	1238.5 M

**Table 3 nutrients-16-00200-t003:** Performance comparison on Vireo Food–172. EHFR–Net–x: x denotes width multiplier on the base model.

Method	Top-1 Acc.	#Params	#FLOPs
ShuffleNetV2-0.5 [19]	74.3%	0.5 M	41.6 M
**EHFR** **–Net-0.5**	**89.4%**	0.8 M	428.3 M
MobileViTv2-0.5 [39]	87.3%	1.2 M	480.2 M
ShuffleNetV2-1.0 [19]	81.0%	1.4 M	148.9 M
MobileNetV3-0.5 [23]	83.0%	1.6 M	73.4 M
GhostNetV2-0.5 [50]	81.8%	1.8 M	54.1 M
**EHFR** **–Net-0.75**	**91.0%**	1.8 M	956.0 M
MobileViTv2-0.75 [39]	88.0%	2.5 M	1051.4 M
ShuffleNetV2-1.5 [19]	82.4%	2.7 M	303.7 M
**EHFR** **–Net-1.0**	**91.3%**	2.8 M	1210.3 M
MobileNetV3-0.75 [23]	85.9%	2.9 M	162.0 M
MobileNetV3-1.0 [23]	86.7%	4.4 M	219.0 M
**EHFR** **–Net-1.25**	**91.7%**	4.5 M	2066.5 M
MobileViTv2-1.0 [39]	88.2%	4.5 M	1843.4 M
EfficientNeT B0 [21]	83.6%	4.8 M	567.0 M
GhostNetV2-1.0 [50]	84.7%	5.1 M	117.0 M
ShuffleNetV2-2.0 [19]	83.8%	5.7 M	596.6 M
**EHFR** **–Net-1.5**	**91.8%**	6.5 M	2985.5 M
MobileNetV3-1.25 [23]	86.9%	6.5 M	366.9 M
MobileViTv2-1.25 [39]	87.9%	6.9 M	2856.1 M
GhostNetV2-1.3 [50]	85.7%	7.9 M	282.5 M
MobileNetV3-1.5 [23]	86.5%	8.7 M	500.4 M
EHFR–Net-1.75	91.7%	8.8 M	3891.4 M
MobileViTv2-1.5 [39]	88.6%	10 M	4089.4 M
**EHFR** **–** **Net-2.0**	**91.9%**	11.1 M	4787.5 M
GhostNetV2-1.6 [50]	86.2%	11.3 M	415.1 M
MobileViTv2-1.75 [39]	89.1%	13.5 M	5543.5 M
GhostNetV2-1.9 [50]	86.0%	15.4 M	572.9 M
MobileViTv2-2.0 [39]	89.4%	17.6 M	7218.4 M

**Table 4 nutrients-16-00200-t004:** Performance comparison on UEC Food256. EHFR–Net–x: x denotes width multiplier on the base model.

Method	Top-1 Acc.	#Params	#FLOPs
ShuffleNetV2-0.5 [19]	74.3%	0.5 M	41.6 M
**EHFR** **–Net-0.5**	**89.4%**	0.8 M	428.3 M
MobileViTv2-0.5 [39]	69.1%	1.2 M	465.9 M
ShuffleNetV2-1.0 [19]	55.2%	1.5 M	149.0 M
MobileNetV3-0.5 [23]	62.1%	1.7 M	73.5 M
**EHFR** **–Net-0.75**	**71.5%**	1.8 M	956.0 M
GhostNetV2-0.5 [50]	61.1%	1.9 M	54.2 M
MobileViTv2-0.75 [39]	69.8%	2.6 M	1051.5 M
ShuffleNetV2-1.5 [19]	57.5%	2.7 M	303.7 M
**EHFR** **–Net-1.0**	**71.6%**	2.9 M	1210.4 M
MobileNetV3-0.75 [23]	64.9%	3.0 M	162.1 M
MobileNetV3-1.0 [23]	65.5%	4.5 M	219.0 M
MobileViTv2-1.0 [39]	70.0%	4.5 M	1843.4 M
**EHFR** **–Net-1.25**	**71.9%**	4.6 M	2066.6 M
EfficientNet B0 [21]	64.0%	4.9 M	567.1 M
GhostNetV2-1.0 [50]	63.9%	5.2 M	177.1 M
ShuffleNetV2-2.0 [19]	60.1%	5.9 M	596.7 M
EHFR–Net-1.5	72.3%	6.5 M	2941.3 M
MobileNetV3-1.25 [23]	65.7%	6.6 M	367.0 M
MobileViTv2-1.25 [39]	71.2%	7.0 M	2856.1 M
GhostNetV2-1.3 [50]	65.0%	8.0 M	282.7 M
**EHFR** **–Net-1.75**	**72.6%**	8.8 M	3840.2 M
MobileNetV3-1.5 [23]	67.1%	8.8 M	500.5 M
MobileViTv2-1.5 [39]	71.2%	10.0 M	4089.5 M
**EHFR** **–Net-2.0**	**72.7%**	11.2 M	4731.2 M
GhostNetV2-1.6 [50]	65.5%	11.4 M	415.2 M
MobileViTv2-1.75 [39]	71.4%	13.6 M	5543.6 M
GhostNetV2-1.9 [50]	66.1%	15.5 M	573.0 M
MobileViTv2-2.0 [39]	71.5%	17.7 M	7218.5 M

**Table 5 nutrients-16-00200-t005:** Ablation study on Food-101, Food-172, and Food256. MHA: Multi-Head Attention, ANS: Adjusted Network Structure, TNS: Traditional Network Structure.

Dataset	Ablation	Top-1 Acc.	#Params	#FLOPs
Food-101	**EHFR**–**Net-1.0**	**90.67%**	2.82 M	1210.31 M
	LP–ViT → MHA	88.80%	3.67 M	6292.49 M
	w/o LP–ViT	88.37%	3.19 M	1082.62 M
	w/o CNN	87.81%	2.89 M	1667.64 M
	ANS → TNS	87.95%	4.52 M	842.1 M
Food-172	**EHFR**–**Net-1.0**	**91.38%**	2.84 M	1210.33 M
	LP–ViT → MHA	89.66%	3.70 M	6292.51 M
	w/o LP–ViT	90.05%	3.21 M	1082.64 M
	w/o CNN	89.86%	2.91 M	1667.67 M
	ANS → TNS	89.36%	4.55 M	842.15 M
Food256	**EHFR**–**Net-1.0**	**71.30%**	2.87 M	1210.36 M
	LP–ViT → MHA	69.28%	3.72 M	6292.54 M
	w/o LP–ViT	68.27%	3.24 M	1082.67 M
	w/o CNN	68.19%	2.94 M	1667.69 M
	ANS → TNS	67.26%	4.57 M	842.18 M

## Data Availability

Our code is available at https://github.com/LduIIPLab/CVnets.git, accessed on 4 January 2024.

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
