# Peer review of "A Lightweight Hybrid Model with Location-Preserving ViT for Efficient Food Recognition"

_nutrients, 2024, doi:10.3390/nu16020200_

Round 1

Reviewer 1 Report

Comments and Suggestions for Authors

A manuscript titled 'A Lightweight Hybrid Model with Location-Preserving ViT for Efficient Food Recognition' proposes a novel method for classifying food images using a lightweight yet efficient model.

Comments:

The authors have conducted a thorough analysis, and their results and methods are impressive. However, the presentation of the manuscript makes it challenging to follow.

a. The author should consider merging some parts of the 'related work' section with the introduction.

b. A discussion section is missing; the author should include it to highlight key points based on the "related works" section. Also, improvements and limitations of the current work should be discussed in the discussion section based on a thorough comparison of previous studies mentioned in the "related works" section.

c. The author did not include limitations of the current work, which should be addressed in the discussion section.

d. A detailed description of the dataset is missing. Does the dataset belong to a particular country or food type, or does it represent global food?

e. The overall quality of English is acceptable, but in some places, sentences use 'and,' making them long and difficult to follow. It is advised for the author to proofread the manuscript and simplify the sentences.

f. Authors should relocate '4. Experiments' to the 'Material and Methods' section and divide the methods and results from the '4. Experiments' into their respective sections, 'Material and Methods' and 'Results.'

g. Additionally, create a distinct section for 'Results' and transfer all results from '4. Experiments' to this newly created section.

h. In addition, Figure 2 is crucial as it provides a summary of the entire work. Please include a description similar to that in Figure-4 or other figures for better clarity.

Comments on the Quality of English Language

The author should improve the quality of English.  The author should improve the quality of English.   The clarity of some sentences is poor and difficult to read. In some places, sentences use 'and,' making them long and difficult to follow. It is advised for the author to proofread the manuscript and simplify the sentences.

Author Response

Thank you very much for your in-depth analysis and the valuable comments on this manuscript.

[Comments-1] The author should consider merging some parts of the 'related work' section with the introduction.

[Response-1]

Thank you for your suggestions. We found that the last paragraph of the "Related Works" section introduces the motivation and overview of our work. As the reviewer pointed out, this content has either been covered in the introduction section or is more appropriate for the introduction section. Therefore, we have deleted this paragraph and integrated some of its content into the introduction section.

[Comments-2]. A discussion section is missing; the author should include it to highlight key points based on the "related works" section. Also, improvements and limitations of the current work should be discussed in the discussion section based on a thorough comparison of previous studies mentioned in the "related works" section.

[Response-2]

Thank you for your suggestions. In the final section of "Results," a supplementary sub-section labeled "Discussion" has been appended, wherein a thorough discussion is provided. Addressing the three major categories of approaches delineated in the "Related Work" section, we discuss their respective strengths and limitations, elucidating the specific advantages of our approach. Simultaneously, an analysis of the limitations inherent in our approach is presented, along with directions for further improvement in the subsequent steps.

[Comments-3] The author did not include limitations of the current work, which should be addressed in the discussion section.

[Response-3]

Thank you for your suggestions. We have discussed the limitations of our work in the newly added Section 4.6, "Discussion."

[Comments-4] A detailed description of the dataset is missing. Does the dataset belong to a particular country or food type, or does it represent global food?

[Response-4]

Thank you for your suggestions. In Section 3 "Materials and Methods", Subsection 3.1  "Datasets and Training settings", we have incorporated detailed descriptions of the three datasets, specifying the countries and regions represented in each dataset, the total number of food categories covered, and the overall count of images.

[Comments-5] The overall quality of English is acceptable, but in some places, sentences use 'and,' making them long and difficult to follow. It is advised for the author to proofread the manuscript and simplify the sentences.

[Response-5]

Thank you for your suggestions. Following your suggestions, we have rephrased a total of nine lengthy and complex sentences throughout the manuscript. They are highlighted for your review in the text.

[Comments-6] Authors should relocate '4. Experiments' to the 'Material and Methods' section and divide the methods and results from the '4. Experiments' into their respective sections, 'Material and Methods' and 'Results.'

[Response-6]

Thank you for your suggestions. Following your recommendation, we have divided Section '4. Experiments' with a portion integrated into Section 'Material and Methods' and another part reorganized into Section 'Results'.

[Comments-7] Additionally, create a distinct section for 'Results' and transfer all results from '4. Experiments' to this newly created section.

[Response-7]

Thank you for your suggestions. Following your suggestion, we have established a separate Section 'Results', and transferred all experimental results from '4. Experiments' to this newly created section.

[Comments-8] In addition, Figure 2 is crucial as it provides a summary of the entire work. Please include a description similar to that in Figure-4 or other figures for better clarity.

[Response-8]

Thank you for your suggestions. Following your suggestion, we have revised and enriched the description of Figure 2. Each part of the processing workflow in the figure has been detailed, allowing for a comprehensive understanding of the entire workflow through a thorough examination of the image.

Reviewer 2 Report

Comments and Suggestions for Authors

Sheng et al. reported a cyber lightweight system for food recognition.  My detailed suggestions are stated below:

1.       Line 21, please rephrase “skip connection” to clarify the meaning.

2.       Line 25, for readers that are not familiar with these food recognition databases, I think it is necessary to give ETHA Food-101 a reference or brief introduction.

3.       Line 28, is there a reason this sentence is bolded?

4.       Line 32-33, revise the English to convey a meaningful idea please. 

5.       Line 112, please specify the three open-source food image recognition databases

6.       Through the manuscript, the authors respectively used the word dataset?  Are these datasets or databases?

Comments on the Quality of English Language

Minor revision recommended

Author Response

Thank you very much for your constructive and valuable comments.

[Comments-1] Line 21, please rephrase “skip connection” to clarify the meaning.

[Response-1]

Thank you for your suggestions. We employed a more detailed descriptive approach to articulate the specific meaning of the "skip connection" operation. We replaced “skip connection” by “directly summing and concatenating the outputs from the convolutional and ViT”.

[Comments-2] Line 25, for readers that are not familiar with these food recognition databases, I think it is necessary to give ETHA Food-101 a reference or brief introduction.

[Response-2]

Thank you for your suggestions. In Section 3 "Materials and Methods", Subsection 3.1  "Datasets and Training settings", we have incorporated detailed descriptions of the three datasets, specifying the countries and regions represented in each dataset, the total number of food categories covered, and the overall count of images.

[Comments-3] Line 28, is there a reason this sentence is bolded?

[Response-3]

Thank you for your suggestions. Our intention was to convey that we would provide the source code for the entire work. We have now removed the bold formatting and provided a link to the source code.

[Comments-4] Line 32-33, revise the English to convey a meaningful idea please.

[Response-4]

Thank you for your suggestions. We rewrite the sentence as this: ‘Food computing has gained increased attention in the fields of food and nutrition[1]. This is due to its potential to contribute to improvements in diet, health, and the food industry[2-5]’.

[Comments-5] Line 112, please specify the three open-source food image recognition databases

[Response-5]

Thank you for your suggestions. In Section 3 "Materials and Methods", Subsection 3.1  "Datasets and Training settings", we have incorporated detailed descriptions of the three datasets, specifying the countries and regions represented in each dataset, the total number of food categories covered, and the overall count of images.

[Comments-6] Throughout the manuscript, the authors respectively used the word dataset?  Are these datasets or databases?

[Response-6]

Thank you for your suggestions. The intended meaning is likely "dataset." We have checked and found that we mistakenly wrote one instance as "database", and it has been corrected to "dataset."

Reviewer 3 Report

Comments and Suggestions for Authors

In the article - a novel Lightweight Hybrid Model is proposed on the basis of Efficient Hybrid Food Recognition Net(EHFR-Net), a novel neural network that integrates Convolutional Neural Networks(CNN) and Vision Transformer(ViT). The model is designed to capture the specific characteristics of food imaging, enabling capability of empowering individuals to effectively manage their daily diet and nutrition using devices such as smartphones. However the topic of the manuscript is not relevant for the Journal, taking into consideration that evaluation of such models could be more suitably assessed in Scientific Journals in field of Mathematics and Informatics or Computer Science.  Moreover the References provided are namely in such a Journals [e.g. only References 6 and 7 are covering the scope of Food and Nutrition Science]. Nevertheless, the work has a very high application potential in the field of Nutrition studies, and once the Model is validated its application for analyzing factors such as the meal type, components, and other distinctive information, for  evaluation of  the nutritional value of a meal and understand individual dietary habits.

Author Response

Thank you very much for the valuable and detailed comments.

[Comments-1] In the article - a novel Lightweight Hybrid Model is proposed on the basis of Efficient Hybrid Food Recognition Net(EHFR-Net), a novel neural network that integrates Convolutional Neural Networks(CNN) and Vision Transformer(ViT). The model is designed to capture the specific characteristics of food imaging, enabling capability of empowering individuals to effectively manage their daily diet and nutrition using devices such as smartphones. However the topic of the manuscript is not relevant for the Journal, taking into consideration that evaluation of such models could be more suitably assessed in Scientific Journals in field of Mathematics and Informatics or Computer Science.  Moreover the References provided are namely in such a Journals [e.g. only References 6 and 7 are covering the scope of Food and Nutrition Science]. Nevertheless, the work has a very high application potential in the field of Nutrition studies, and once the Model is validated its application for analyzing factors such as the meal type, components, and other distinctive information, for  evaluation of  the nutritional value of a meal and understand individual dietary habits.

[Response-1]

Thank you for your suggestions. We submitted to the Special Issue "Mobile Health and Nutrition (2nd Edition)," and the first keyword in the listed scope of submitted papers on the special issue's homepage is food recognition. Considering that the mainstream technology for food recognition is currently based on deep learning, our work focuses on lightweight processing of deep learning-based food recognition. This is aimed at facilitating the practical implementation of this technology on the application end such as mobile phones. Therefore, we think that our work aligns with the theme of this special issue.

Round 2

Reviewer 1 Report

Comments and Suggestions for Authors

This is the revised version of the manuscript. The authors have addressed all the concerns raised during the previous review. 

Comments on the Quality of English Language

I believe there are areas in the manuscript where the flow and clarity of sentences could be improved. I would suggest that the authors thoroughly proofread the manuscript.

Reviewer 3 Report

Comments and Suggestions for Authors

In the article - a novel Lightweight Hybrid Model is proposed on the basis of Efficient Hybrid Food Recognition Net(EHFR-Net), a novel neural network that integrates Convolutional Neural Networks(CNN) and Vision Transformer(ViT). The model is designed to capture the specific characteristics of food imaging, enabling capability of empowering individuals to effectively manage their daily diet and nutrition using devices such as smartphones.

Taking into consideration that the article is submitted to the Special Issue "Mobile Health and Nutrition (2nd Edition)," focusing on food recognition, the manuscript is suitable for the Journal.

The related works in the field are systematically referred and the methods applied are precisely described. The results demonstrate the effectiveness if the model and the contribution of the work is well summarized.

The work has a very high application potential in the field of Nutrition studies for analyzing factors such as the meal type, components, and other distinctive information, for evaluation of the nutritional value of a meal and understand individual dietary habits and could be proposed for publishing the Nutrients Journal in the present form.